# Identification of Candidate QTLs and Genes for Ear Diameter by Multi-Parent Population in Maize

**DOI:** 10.3390/genes14061305

**Published:** 2023-06-20

**Authors:** Fuyan Jiang, Li Liu, Ziwei Li, Yaqi Bi, Xingfu Yin, Ruijia Guo, Jing Wang, Yudong Zhang, Ranjan Kumar Shaw, Xingming Fan

**Affiliations:** 1Institute of Food Crops, Yunnan Academy of Agricultural Sciences, Kunming 650205, China; jiangfuyansxx@126.com (F.J.); liuliyaas@163.com (L.L.); biyq122627@163.com (Y.B.); xingfuyin626@163.com (X.Y.); sherrygrj@163.com (R.G.); jingwang1125@163.com (J.W.); mikezhangy@yahoo.com (Y.Z.); ranjanshaw@gmail.com (R.K.S.); 2Yunnan Dehong Dai and Jingpo Nationality Institute of Agricultural Sciences, Mangshi 678400, China; liziwei9898@163.com

**Keywords:** maize, ear diameter, candidate gene, GWAS, QTL

## Abstract

Ear diameter (ED) is a critical component of grain yield (GY) in maize (*Zea mays* L.). Studying the genetic basis of ED in maize is of great significance in enhancing maize GY. Against this backdrop, this study was framed to (1) map the ED-related quantitative trait locus (QTL) and SNPs associated with ED; and (2) identify putative functional genes that may affect ED in maize. To accomplish this, an elite maize inbred line, Ye107, which belongs to the Reid heterotic group, was used as a common parent and crossed with seven elite inbred lines from three different heterotic groups (Suwan1, Reid, and nonReid) that exhibited abundant genetic variation in ED. This led to the construction of a multi-parent population consisting of 1215 F_7_ recombinant inbred lines (F_7_RILs). A genome-wide association study (GWAS) and linkage analysis were then conducted for the multi-parent population using 264,694 high-quality SNPs generated via the genotyping-by-sequencing method. Our study identified a total of 11 SNPs that were significantly associated with ED through the GWAS, and three QTLs were revealed by the linkage analysis for ED. The major QTL on chromosome 1 was co-identified in the region by the GWAS at SNP_143985532. SNP_143985532, located upstream of the *Zm00001d030559* gene, encodes a callose synthase that is expressed in various tissues, with the highest expression level in the maize ear primordium. Haplotype analysis indicated that the haplotype B (allele AA) of *Zm00001d030559* was positively correlated with ED. The candidate genes and SNPs identified in this study provide crucial insights for future studies on the genetic mechanism of maize ED formation, cloning of ED-related genes, and genetic improvement of ED. These results may help develop important genetic resources for enhancing maize yield through marker-assisted breeding.

## 1. Introduction

Ear diameter (ED) is one of the most important components of grain yield (GY) in maize, and understanding its genetic base will facilitate increasing GY, which is a major breeding goal for maize. Maize is an important source of cooking oil, biofuel, and animal feed. By 2050, the predicted 9 billion people in the world will require 70% more food than today’s population [1].

As a typical quantitative genetic trait, ED is regulated by multiple genes and greatly impacted by environmental factors [2,3]. QTL mapping has been used as an effective method and an ideal tool for identifying quantitative trait loci (QTL). Zhang et al. (2010) located five ED-related QTLs (*Qednr9-1*, *Qednr9-2*, *Qedns4-1*, *Qedns4-2*, and *Qedns9-1*) on chromosomes 4 and 9 in high and low nitrogen environments, respectively, using a segregating population consisting of 239 recombinant inbred lines (RILs) at the ninth generation or F_9_RILs [4]. Yu et al. (2014) reported a total of six ED-related QTLs using an F_2_ segregating population from a sweet maize inbred line and its space-induced mutant [5]. Yang et al. (2015) detected two ED-related QTLs (*q10sED8-1* and *q10sED9-1*) on chromosomes 8 and 9, respectively, using a segregating population of 233 families derived from maize inbred lines B73 and SICAU1212 [6]. Mendes-Moreira et al. (2015) discovered 10 ED-related QTLs using an F_2_ population derived from maize inbred lines PB260 and PB266 [7]. Su et al. (2017) identified five ED-related QTLs (*qEAD-1*, *qEAD-2*, *qEAD-3*, *qEAD-4*, and *qEAD-5*) on chromosomes 1, 4, and 7, using a segregating population of 199 F_2_ individuals obtained by crossing maize inbred lines SG-5 and SG-7 [8]. Although all the above studies detected some ED-related QTLs, only a few of them identified putative genes responsible for maize ED, due to the large QTL intervals obtained with segregating populations only.

A genome-wide association study (GWAS) is a powerful tool for analyzing the genetic structure of complex traits and overcomes the limitations of QTL mapping by narrowing down candidate regions. In recent years, GWAS has been successfully used to identify QTLs and candidate genes for maize kernel-related traits, providing valuable insights into maize genetic architecture [9,10,11,12,13]. However, GWAS has its weakness; it frequently suffers from the “missing heritability” problem, i.e., the correlation between the phenotype of complex traits and underlying markers cannot fully explain the phenotypic variations [14]. Studies have shown that rare variants and variants with low minor allele frequency may be responsible for the missing heritability of complex traits [15].

To overcome spurious associations and increase the detection power of rare alleles in crop species, multi-parent populations using different cross designs have been developed [16]. Among these, one type of multi-parent population, the nested association mapping (NAM) population, has advantages over bi-parental populations. It has a lower sensitivity to genetic heterogeneity, can produce additional recombination breakpoints, and increases allelic diversity and the power of QTL detection [17]. The NAM population was first proposed in maize. It consisted of a set of 25 RIL families derived from crosses between the maize inbred line B73 and 25 genetically diverse inbred lines representing the domesticated maize gene pool worldwide [18]. This population has been utilized for genetic analysis of complex quantitative traits such as yield-related traits, leaf morphology, flowering time, starch, protein, oil, and carbon/nitrogen (C/N) metabolism [17,19,20,21]. The Bauer laboratory in Europe then constructed two sets of European-NAM (EU-NAM) populations using European Dent and Flint maize germplasms to analyze the genetic basis of maize [22]. Due to their powerful function in dissecting the genetic architecture of complex traits, NAM populations have been widely applied in various crops such as rice, wheat, barley, rapeseed, soybean, and peanut [23,24,25,26,27,28,29]. The above studies have fully utilized the advantages of NAM populations, including abundant recombination between parents and families, and large genetic differences, which to some extent eliminated the negative effects of population structure.

To gain a better understanding of the genetic mechanism behind maize ED, and to identify significant QTLs, SNPs, and important candidate genes, seven elite inbred lines (TRL02, CML373, CML312, CML395, Q11, D39, and Y32), with abundant genetic variations in ED from the Suwan1, Reid, and nonReid heterotic groups, were used as donor parents and crossed with an elite maize inbred line, Ye107, from the Reid heterotic group. A multi-parent population consisting of 1215 F_7_RILs was constructed from the above crosses [30]. This multi-parent population was then subjected to QTL mapping and genome-wide association analysis. The objectives of this study were to (1) identify significant SNPs and QTLs associated with ED; (2) perform co-localization analysis by combining GWAS and linkage mapping to further identify the candidate genes, and (3) investigate the applicability of the “three heterotic group” pattern for improving ED in maize breeding.

## 2. Materials and Methods

### 2.1. Selection of Parental Lines for Constructing the Multi-Parent Population

Our team proposed a “three heterotic group” pattern consisting of Suwan1xReid, Suwan1×nonReid, and ReidxnonReid, based on long-term breeding practice [31,32,33]. In this study, a multi-parent population consisting of 1215 F_7_RIL families was constructed by utilizing an elite maize inbred line, Ye107 (male), from the Reid heterotic group as a common parent crossed with seven elite female inbred lines: D39 (Suwan1), Y32 (Suwan1), Q11 (Reid), TRL02 (nonReid), CML373 (nonReid), CML312 (nonReid), and CML395 (nonReid) from the three heterotic groups, respectively.

These inbred lines were purposely selected based on their heterotic group classification, significant differences in ED (*p* < 0.001), and combining abilities, as well as heterosis in actual breeding practice [30,34]. Figure 1 shows an example of the crosses where the inbred lines D39 (Suwan1), Ye107 (Reid), and TRL02 (nonReid) were used as parents, resulting in the development of three excellent maize hybrids with large ED. These hybrids have been widely promoted and applied. Based on this example, we have strong confidence that a multi-parent population constructed from these parents should facilitate the detection of ED-related QTLs and functional genes. It will further validate the significance and practical breeding value for improving ED and GY in maize using the “three heterotic group” pattern.

### 2.2. Field Experiment and Phenotypic Data Collection

The multi-parent population, consisting of 1215 maize F_7_RILs and 8 parents, was planted in Dehong (longitude: 98.6° E, latitude: 24.4° N) and Baoshan (longitude: 99.2° E, latitude: 25.1° N) in Yunnan Province, China in 2019 using a randomized complete block design (RCBD) with three replications at each location. Each F_7_RIL was planted in a two-row plot with row spacing of 70 cm and 14 plants per row, resulting in a plant density of approximately 62,112 plants per hectare. The detailed methods of measuring ED are as follows:

Ten ears were consecutively harvested from 10 plants of each F_7_RIL. The width of the mid-section of each drying ear was measured using a vernier caliper, and the mean ED of the ten ears was recorded as the final ED (cm) of each F_7_RIL.

The collected data was analyzed using the best linear unbiased prediction (BLUP) method with the lme4 package in R (v3.2.2). The analysis was performed using the following formula:

Multi-locations:$$Y_{ijlk}=\mu +{Line}_{i}+{Loc}_{j}+{\left(Line\times Loc\right)}_{ij}+\mathrm{Rep}\;{\left(Loc\right)}_{jl}+\varepsilon_{ijlk}$$

One-location:$$Y_{ilk}=\mu +{Line}_{i}+\varepsilon_{ilk}$$
where *Y_ijlk_*, *μ*, *Line_i_*, and *Loc_l_* represent the ED phenotype values of each ear, intercept, *i*th line effects, and *l*th location effects, respectively. Rep*_j_* represents the *j*th replication effect, and *ε_ijlk_* represents the random effects. (*Line × Loc*)*_ij_* is used to display the interaction of the *i*th line at the *j*th location, and Rep (*Loc*)*_jl_* shows the nested effect of the *j*th replication within the *l*th location. The $Y_{ilk}$ is the ED phenotype values of each ear at one location, and $\varepsilon_{ilk}$ represents the random effects.

Mid-parent heterosis (MPH) was determined as follows:MPH = [(F1 − MP)/MP] × 100
where F1 = mean of hybrid, MP = mid-parent value = (P1 + P2)/2, and where P1 = Parent1 and P2 = Parent2

### 2.3. Genotyping-by-Sequencing (GBS)

The genomic DNA was extracted from the seedling leaves of each F_7_RIL using the cetyl trimethyl ammonium bromide (CTAB) method [35]. Then, the isolated genomic DNA from each F_7_RIL was digested with the restriction endonucleases PstI and MspI (New England BioLabs, Ipswich, MA, USA) and ligated with barcode adapters using T4 ligase (New England BioLabs). GBS DNA libraries were constructed and sequenced following the GBS protocol [36].

All the ligated samples were pooled and purified using the QIAquick PCR Purification Kit (QIAGEN, Valencia, CA, USA). Primers complementary to both adaptors were used for polymerase chain reaction (PCR) amplication. The PCR products were then purified and quantified with the Qubit dsDNA HS Assay Kit (Life Technologies, Grand Island, NY, USA). After selecting PCR products of 200–300 bp size in an Egel system (Life Technologies), the library concentration was estimated using a Qubit 2.0 fluorometer and the Qubit dsDNA HS Assay Kit (Life Technologies). The sequencing was carried out using an Ion Proton sequencer (Life Technologies, software version 5.10.1) with P1v3 chips. The final reads were generated using TASSEL v5.0 (https://github.com/Euphrasiologist/GBS_V2_Tassel5, accessed on 6 March 2022) [37]. Prior to the TASSEL analysis, 80 poly (A) bases were appended to the 3′ ends of all the sequencing reads. The SNPs were called using the Genome Analysis Toolkit software [38] with the maize B73 reference genome (B73_V4, ftp://ftp.ensemblgenomes.org/pub/plants/release-37/fasta/zea_mays/DNA, accessed on 6 March 2022) [39]. In total, 264,694 high-quality SNPs were generated and annotated using the ANNOVAR software tool (v2013-05-20) [40].

### 2.4. Genome-Wide Association Studies (GWAS)

The GWAS was conducted using the efficient mixed-model association (EMME) analysis method in the GEMMA (genome-wide efficient mixed-model association) software package [41]. The following mixed-model analysis was used for the GWAS analysis:y = Xa + Sb + Km + e;
where y represents phenotype, a and b are fixed effects that represent marker effects and non-marker effects, respectively, and m represents unknown random effects. The incidence matrices for a, b, and m are represented by X, S, and K, respectively, while e is a vector of the random residual effects. To correct for population structure, we used the top three principal components (PCs) to build the S matrix, while the kinship (K) matrix was built using the matrix of simple matching coefficients. We modeled the genetic relationship between individuals as a random effect using the K matrix. Significant *p*-value thresholds (*p* < 1 × 10^−6^) were set to control the genome-wide type 1 error rate.

We used PLINK [42] to calculate the independent marker with the parameters (--indep-pairwise 50 5 0.2). A significance threshold of −log10(P) > 4.5, calculated using the formula −log10(1/SNP numbers), was used to identify significant SNPs associated with maize ED. Candidate genes related to ED, located within the 50 Kb flanking regions (LD area) of the peak SNPs, were identified and annotated using the B73 reference genome. Since the calculated LD based on GBS data may not be highly accurate, we used an empirical value in combination with the Haploview result to accurately define LD blocks within specific intervals. In maize, the typical LD length for an inbred line ranges from 10 to 100 kb. In our analysis, we selected a conservative interval size of 50 kb, which is a middle value within this range, for the purpose of LD block determination.

### 2.5. Linkage Mapping and QTL Analysis

The subpopulation Y32×Ye107 and its 162 F_7_RILs families were selected from the multi-parent population for genetic linkage analysis. We have analyzed all the other populations and phenotypes. Finally, we carefully selected the population and phenotypes with the best effect to display the results on the map. We defined the best results based on whether the SNPs identified by GWAS and QTL by linkage analysis were present at the same location. The Ye107/Y32 was the only population where the SNPs and QTL overlapped. Allelic SNPs were then utilized to construct genetic linkage maps through the JoinMap 4 software [43]. The linkage groups were formed using a LOD threshold of ≥5.0. QTLs for ED were identified using the composite interval mapping (CIM) method via Windows QTL Cartographer v2.0 [44]. The LOD threshold was set based on 1000 random permutation tests with a significance level of *p* ≤ 0.05 [45]. QTLs with a LOD threshold of ≥2.5 were considered significant. The percentage of phenotypic variation explained (PVE) by individual QTLs was calculated by the square of the partial correlation coefficient (R^2^).

### 2.6. Analysis of Epistatic Effect of SNPs

With the dataset obtained from GWAS, and with SNPs that were significantly associated (−log10(P) >= 4.5), we utilized the PLINK software v1.9 [41] to calculate the epistatic effects of pairwise SNPs. The optimal epistatic SNPs for each SNP were then determined using the following parameters:/plink-1.07-x86_64/plink --file test --pheno phenoq.txt --epistasis --set-test --set epi.set2 --epi1 1 --epi2 0.3 --noweb --out testset1set2.

## 3. Results

### 3.1. Phenotypic Evaluation for ED

The eight parents of the multi-parent population exhibited large variations in ED, ranging from 3.4 to 4.6 cm. The common parent, Ye107, had the smallest ED, while the seven donor parents had statistically significantly larger ED (*p* < 0.01) than Ye107. Subsequently, the F_7_RILs families of the seven subpopulations showed significant differences in ED (Figure 2A,C,D). The coefficient of variation (CV) analysis revealed that the ED variations of the seven subpopulations ranged from 9.1% (CML373) to 15.0% (CML312) in Dehong and from 9.0% (CML373) to 13.7% (CML312) in Baoshan (Table 1). To further assess the reliability of the phenotypic identification, we calculated the correlation coefficient between the two environments, and the result demonstrated a highly significant positive correlation (r = 0.899, *p* < 0.0001) for the ED phenotype. Overall, the multi-parent population exhibited wide variations for ED; however, the variation of ED was consistent between the two environments, indicating high reliability of the phenotypic data for further analysis.

### 3.2. Genome-Wide Association Analysis for ED

The ED of the multi-parent population showed significant differences between the two environments by paired *t*-test (*p* = 0.027) in Dehong and Baoshan and tended to follow a normal distribution (Figure 3). In order to conduct a GWAS analysis on ED in these two environments, the MLM model was employed, based on the 264,694 high-quality SNPs and the ED phenotypes of the 1215 F_7_RILs families of the multi-parent population. A threshold of −log10(P) > 4.5 was set, and 11 SNPs were identified to be significantly associated with ED, 6 in Dehong and 5 in Baoshan (Figure 3). These SNP loci were located on chromosomes 1, 2, 3, 6, 8, and 9 (Table 2), with two SNPs located on chromosomes 1 and 6 being detected in both environments (Figure 3).

### 3.3. Linkage Analysis for ED and the Identification of ED-Related Genes

In this study, the linkage map from the Ye107/Y32 subpopulation was used to identify significant QTLs for ED in Dehong (DH). One significant QTL was detected on chromosome 1, explaining 7.1% of the ED phenotypic variation. Additionally, two significant QTLs were detected on chromosome 9, explaining 9.2% and 10.5% of the phenotypic variation, respectively, for ED (Figure 4A and Table 3). Notably, the major QTL on chromosome 1 was co-localized in the region identified by GWAS analysis. Furthermore, the SNP (SNP_143985532) on chromosome 1 showed the highest −log10(p) value and was detected in both environments as well as by BLUP (Figure 4B). Based on the most significant SNP and linkage disequilibrium block analysis, we estimated an ED-related candidate region containing this SNP to range from 143.965 Mb to 144.005 Mb (40 Kb) on chromosome 1 (Figure 4B). Within this 40 kb genomic region, three putative protein-encoding genes were found, and SNP_143985532 was located upstream of the gene *Zm00001d030559*, which encodes callose synthase 1 (Figure 4C). The *Zm00001d030559* gene was found to be expressed at low levels in various tissues [46], with the highest expression level in the ear primordium in maize (Figure 4D). It is well known that the development of maize ear primordium is one of the important factors affecting the development of maize ears. The haplotype analysis showed that there were two haplotypes (HapA and HapB) for the *Zm00001d030559* gene. HapA, carrying the allele ‘GG’, showed significantly shorter ED than HapB, carrying the allele ‘AA’ in both environments and in BLUP for the multi-parent population (Figure 4E). The proportion of HapA in the seven F_7_RIL families ranged from 1.75% to 43.6%, with CML312 having the lowest ratio of HapB, and Y32 having the highest ratio of HapB (Figure 4F). The proportion of HapB was positively correlated with ED, and all the F_7_RIL families (except Ye107/CML312) demonstrated similar relationships between the haplotype and ED (Figure 4F,G). These results suggested that the two SNP_143985532-based haplotypes (HapA and HapB) were statistically associated with ED. Therefore, *Zm00001d030559* is related to ED variation and is an ED-related candidate gene in maize.

### 3.4. Analysis of Epistatic Effect of SNP

Epistasis analysis of significant SNPs in the gene revealed that the SNP (SNP_143985532) identified through both linkage and GWAS analysis exhibited a significant epistasis effect on an SNP (SNP_103192371) from chromosome 5. Further gene mining analysis conducted within 50 kb upstream and downstream of SNP_103192371 identified four genes (*Zm00001d015649*, *Zm00001d015650*, *Zm00001d015651,* and *Zm00001d015653*) in this chromosomal region. Among these genes, *Zm00001d015651* encodes lycopene β-cyclase and chloroplast-specific lycopene β-cyclase, which are involved in carotenoid biosynthesis. Unfortunately, the functions of the other three genes have not been reported yet. These genes represent important candidate genes that may affect ED and warrant further investigation in future studies.

### 3.5. Validation of the Contribution of ED QTLs and Candidate Genes in Determining Grain Yield and Significance of the “Three Heterotic Group” Pattern

The ED, GY, and mid-parent heterosis (MPH) are listed in Table 4, based on the data of eight parents and their F_1_ hybrids. The results show that a cross of Y32×Ye107 had the highest ED, ED_MPH, and GY_MPH of all the crosses. As we know, Y32 was the donor parent of the QTLs identified in this study, suggesting that these QTLs had positive effects on enlarging ED and increasing ED_MPH and GY_MPH. Correlation analysis showed a significant positive correlation between the ED of the parent and hybrid F_1_ (r = 0.830, *p* = 0.021). Additionally, a significant positive correlation was found between ED_MPH and GY_F_1_ (r = 0.776, *p* = 0.04) and between ED_MPH and GY_MPH (r = 0.837, *p* = 0.019). These findings indicate that the SNPs sites, QTLs, and candidate genes of ED carried by these donor parents have positive effects on increasing GY and GY_MPH in maize.

## 4. Discussion

### 4.1. Screening for ED in Maize and Identification of Candidate Genes

Although many QTLs for ED have been detected in recent years [4,5,6,7,8], most of them are not consistent, due to the use of different maize populations in different environments in these studies. Therefore, only a few QTLs and putative candidate genes have been applied in maize breeding, if any.

In this study, we used Ye107 as the common parent, an important maize breeding material in Southwest China, to cross with seven elite maize inbred lines from the Suwan1, Reid, and nonReid heterotic groups. Through this process, we constructed a multi-parent population consisting of 1215 F_7_RILs from seven subpopulations to map ED QTLs and identify functional genes. A total of 11 SNPs significantly associated with ED were detected by GWAS analysis, including 2 SNPs (SNP_143985532 and SNP_157559780) on chromosomes 1 and 6, which were commonly detected in both the Dehong and Baoshan environments (Figure 3 and Table 2).

The SNP_143985532 on chromosome 1 was significantly associated with ED and was co-located by both linkage and GWAS analysis. Further analysis revealed that SNP (SNP_143985532) was located upstream of the gene *Zm00001d030559*, which encodes a callose synthase (Figure 4). Callose is a polysaccharide, widely existing in plant cell walls, which regulates the transport of plasmodesmata and phloem, thereby affecting plant development and response to stress [47].

The major composite of callose is β-1,3-glucan, which is synthesized by callose synthases (*CalS*), also known as *GSL* for glucan synthase-like [47,48]. Previous studies have shown that callose synthases play important roles in various processes in plant growth, development, and stress responses. For instance, in Arabidopsis, *GSL7* mutants display semi-dwarf phenotypes and have small and short flowers, peduncles, and siliques [49].

The mutation of the callose synthase gene *GSL8* delays callose deposition in the cell plates, displays a typical cytokinesis-defective phenotype, and is seedling-lethal in Arabidopsis [50]. In both Arabidopsis and rice, *GSL5* plays a vital role in microspore development, and its mutation results in a severe reduction in fertility [51,52]. In rice, the *crr1* mutant (*AtCalS10* homologous gene) shows a delayed ovary expansion, small grain, and defective vascular cell pattern [53]. In this study, the expression pattern analysis showed that the *Zm00001d030559* gene was expressed at a low level in various tissues, with the highest expression level in maize ear primordium (Figure 4G). These results strongly suggest that this gene is a putative candidate gene controlling ED.

During the vegetative growth stage of maize, the shoot apical meristem (SAM) produces leaves and associated axillary meristems (AMs), which have the potential to develop into ear primordium. As maize transitions from the vegetative to the reproductive phase, the SAM and AM transform into inflorescence meristem (IM), which ultimately develops into tassels and ears. In normal ear development, the IM initially generates indeterminate spikelet pair meristems (SPMs) in the peripheral region. Each SPM produces two spikelet meristems (SMs), and each SM develops into an upper floral meristem (FM) and a lower floral meristem. The lower floral meristem usually aborts, and only the upper floret develops into a kernel [54]. Therefore, AM initiation, IM size, floret patterning, and FM determinacy are critically important for maize ear architecture and GY. Several studies have shown a strong correlation between maize ED and kernel row number (KRN), as well as between ear length and kernel number per row (KNR) [53]. Furthermore, functional analysis of ED genes has also provided evidence supporting these correlations [9,54,55,56,57,58,59,60]. Our study has clearly demonstrated that the Zm00001d030559 gene leads (Figure 4) to a higher expression level in ear primordium than that in other tissues. This result strongly suggests that the gene plays an important function in the development of ear primordium, and it might be an important candidate gene for controlling ED in maize.

Epistatic analysis of the significant SNP_143985532 revealed a significant epistasis effect on the SNP_103192371 on chromosome 5. Further gene mining analysis within 50 kb upstream and downstream of SNP_103192371 identified four genes (*Zm00001d015649*, *Zm00001d015650*, *Zm00001d015651,* and *Zm00001d015653*) in this chromosomal region. Further analysis showed that the gene *Zm00001d015651* encodes lycopene β-cyclase (gene: *crtY*) and chloroplast-specific lycopene β-cyclase (gene: *LCY1*), which are involved in carotenoid biosynthesis. In Arabidopsis, *crtY* catalyzes the double cyclization reaction that converts lycopene to β-carotene and neurosporene to β-zeacarotene, and then converts neurosporene to 7,8-dihydro-β-carotene via monocyclic β-zeacarotene. *LCY1* is involved in improving salt tolerance by increasing the synthesis of carotenoids, which helps mitigate reactive oxygen species (ROS) and protects the photosynthetic system under salt stress [61]. However, the functions of the other three genes have not been reported in previous studies. Therefore, these genes could be important candidate genes that affect ED, and further analysis is necessary to investigate this possibility.

A comprehensive study was conducted by Yang et al. (2020) [62], where they performed GWAS and QTL linkage analysis for three major maize yield-related traits: ear diameter (ED), ear row number (ERN), and kernel number per row (KNR), using 126 inbred lines. They identified 193, 170, and 186 genes from the candidate regions associated with ED, ERN, and KNR, respectively. Additionally, they also discovered nine genes from the co-localized candidate region for both ED and ERN. Notably, they identified the gene *Zm00001d034629* (ts6) from the candidate region of locus chr1-298535881 on chromosome 1, which encodes an AP2-EREBP transcription factor. The gene belongs to the AP2/EREBP family and exhibits higher expression in pre-pollination cob. In our study, we also identified the gene *Zm00001d030559*, located downstream of SNP_143985532 on chromosome 1. This gene appears to be involved in ear primordium formation and is located in a different region on chromosome 1 compared to ts6, discovered by Yang et al., (2020). Hence, we strongly believe that this is a novel candidate gene associated with ED. Another intriguing finding from Yang’s study is the sharing of significant loci by two or more traits. For instance, a significant locus (chr4-190080689) located on chromosome 4 was associated with both ED and ERN. Comparisons with previous studies revealed that the candidate region of the locus (chr1-296366056) is related to both ERN and grain yield, while the candidate region of the locus (chr4-163086906) is associated with ED, KNR, and grain number. These results strongly suggest that candidate genes for one maize yield component trait could also be candidate genes for other yield component traits. Thus, conducting GWAS and QTL linkage analysis for all maize yield components in a single experiment could provide valuable information.

### 4.2. Genetic Effect of ED on “Three Heterotic Group” Pattern

Our research group has proposed a “three heterotic group” pattern for the Suwan1×Reid, Suwan1×nonReid, and Reid×nonReid crosses through long-term breeding practices [31,32,33]. This model offers significant advantages, as there is a large genetic difference between the three populations, allowing for targeted genetic improvements between the two groups without affecting the hybrid advantage between populations. This approach greatly improves breeding efficiency [33], and, using the “three heterotic group” pattern, has resulted in the successful breeding of excellent hybrids [30,31,32,34,63]. Among these hybrids, three excellent hybrids were selected from the multi-parent population parents, namely Yunrui88 (TRL02×Ye107/Reid×nonReid), Desan5 (D39×Ye107/Suwan1×Reid), and Yunrui62 (TRL02×D39/Suwan1×nonReid), all of which exhibited high GY and GY_MPH (Table 4). Their parents belong to the Suwan1, Reid, and nonReid heterotic groups, respectively. This finding further validates the practical applicability of the “three heterotic group” pattern in improving ED and increasing GY in maize breeding and should be used as a guide in maize breeding programs.

Furthermore, our results demonstrated that ED heterosis was significantly positively correlated with GY_F_1_ and GY_MPH, indicating the importance of ED in guiding breeding application research using the “three heterotic group” pattern. These findings have been successfully applied in our breeding practices. For instance, the excellent hybrid variety Yunrui88 (TRL02×Ye107) was approved by Yunnan Province in 2009 and was selected as a leading variety by the Ministry of Agriculture of China in 2014 and 2015. From 2012 to 2014, it was promoted and cultivated in a total area of 350,000 hectares. Additionally, another elite hybrid variety, Yunrui 62, was approved by Yunnan Province in 2017 and passed the national examination and approval by the relevant Chinese authorities in 2020. This hybrid has been promoted and cultivated on approximately 20,000 hectares annually. These results indicate that the “three heterotic group” pattern theory is a feasible guide to deciphering the genetic basis of ED, which could lead to increased GY in future maize breeding programs.

In our study, we conducted whole-genome association mapping and linkage analysis in a multi-parent population, successfully identifying QTLs, SNPs, and candidate genes significantly associated with maize ear diameter. Notably, we discovered a novel gene, *Zm00001d030559*, located downstream of SNP_143985532, which appears to be involved in ear primordium formation. To validate this finding, the expression level of *Zm00001d030559* should be tested in mature ears in future projects. The results obtained from this study not only provide new genetic markers and genomic resources for exploring the genetic structure and molecular mechanisms but also establish a crucial foundation for validating and cloning functional genes related to maize ear diameter.

## Figures and Tables

**Figure 1 genes-14-01305-f001:**
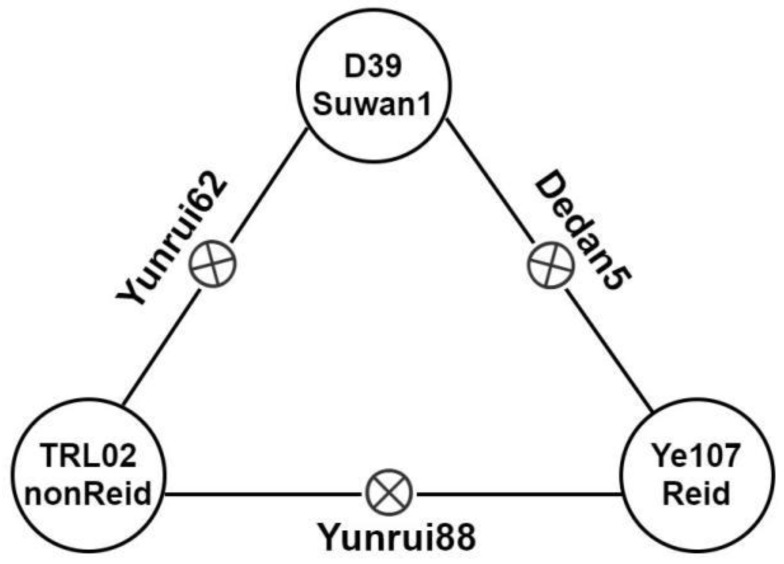
Three representative hybrid varieties were bred, based on the “three heterotic group” pattern theory.

**Figure 2 genes-14-01305-f002:**
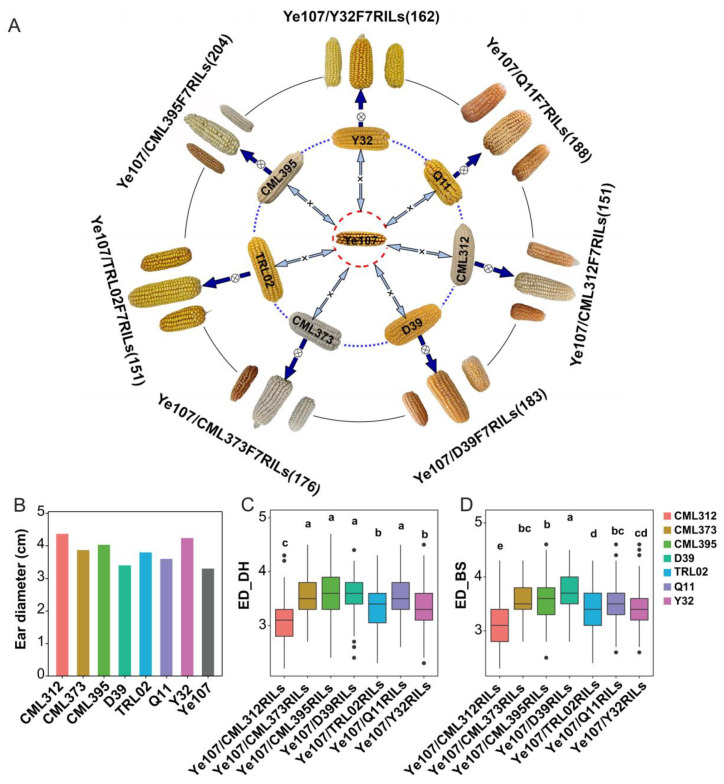
The ear diameter phenotypes of parents and F_7_RIL families of seven subpopulations of a multi-parent population. The three ears in the outer layer are from each RIL family, showing the variations in ED among all RILs in the individual family. The numbers in the brackets represent the numbers of RIL lines in the seven individual subpopulations (**A**); the mean ear diameter data (cm) of eight parents (**B**); mean ear diameters and their variations in the seven subpopulations and Bonferoni test results among the seven subpopulations in Dehong (**C**) and Baoshan (**D**), with the same color scheme used to represent the seven subpopulations as their donor parents. The ear diameters are significantly different at α = 0.05 between RIL subpopulations with different labels of a,b,c,d except between those with same lower case characters.

**Figure 3 genes-14-01305-f003:**
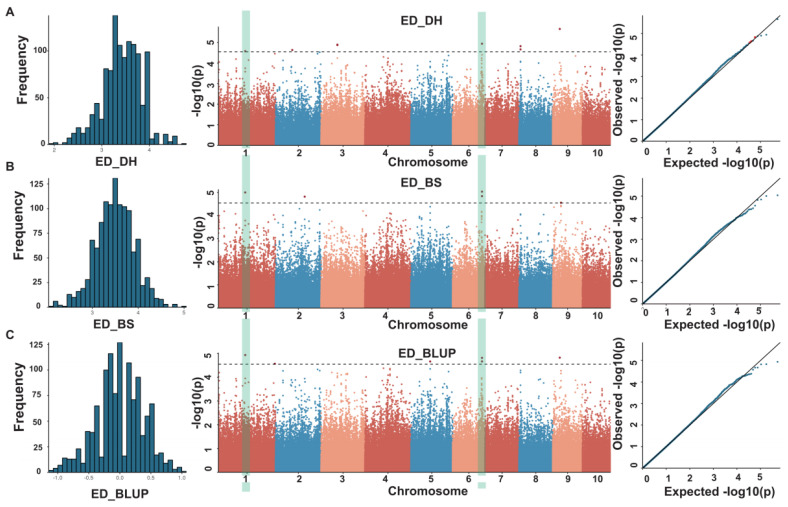
Phenotypic distribution, Manhattan plot, and Q–Q plot for ED traits in Dehong (**A**), Baoshan (**B**), and with BLUP values for all data (**C**).

**Figure 4 genes-14-01305-f004:**
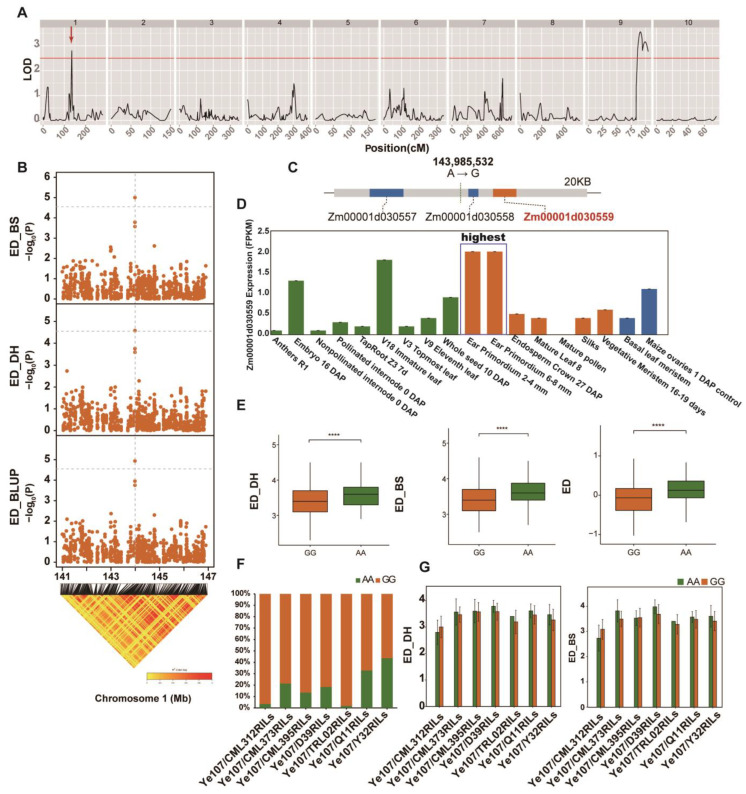
The identification of ED-related genes. All the QTLs detected on different chromosomes of maize in Dehong (DH) in Ye107/Y32 subpopulation (**A**); the position of the most significant SNPs on chromosome 1 identified by GWAS (**B**); the position of candidate genes associated with ED (**C**); the expression levels of the *Zm00001d030559* gene in various tissues, with a higher level observed in ear primordium (**D**); differences in ED between two haplotypes (GG, AA) at two individual locations and overall, with **** indicating *p* < 0.0001 (**E**); the proportions of the two haplotypes observed in seven F7RILs (**F**); the mean ED and 95% range bar for each of the two haplotypes in each F7RIL assessed in the two environments (**G**).

**Table 1 genes-14-01305-t001:** The coefficient of variation (CV) analysis of ear diameter (ED) † phenotypes in Dehong (DH) and Baoshan (BS) environments.

ED_DH	CML312	CML373	CML395	Y32	D39	Q11	TRL02
MIN	2.200	2.700	2.400	2.300	2.400	2.600	2.300
MAX	4.300	4.500	4.700	4.500	4.400	4.500	4.300
mean	3.067	3.560	3.625	3.300	3.602	3.511	3.290
SD	0.460	0.322	0.379	0.384	0.332	0.335	0.425
CV	0.150	0.091	0.105	0.116	0.092	0.095	0.129
**ED_BS**	**CML312**	**CML373**	**CML395**	**Y32**	**D39**	**Q11**	**TRL02**
MIN	2.300	2.800	2.500	2.600	2.800	2.600	2.400
MAX	4.300	4.300	4.600	4.600	4.500	4.600	4.300
mean	3.138	3.569	3.605	3.445	3.725	3.506	3.378
SD	0.429	0.322	0.356	0.359	0.373	0.337	0.413
CV	0.137	0.090	0.099	0.104	0.100	0.096	0.122

†. Ear diameter in cm.

**Table 2 genes-14-01305-t002:** Details of SNPs significantly associated with ED traits.

loc.	chr.	SNP	ref	alt	−log(P)
BS	1	143985532	A	G	5.00
BS	2	156355406	A	G	4.82
BS	6	157559780	C	T	5.04
BS	6	157646040	C	T	4.84
BS	9	44163724	A	G	4.56
DH	1	143985532	A	G	4.58
DH	2	89769384	C	T	4.63
DH	3	87449415	T	G	4.90
DH	6	157559780	C	T	4.94
DH	8	8176464	C	A	4.82
DH	9	37837960	A	T	5.66

**Table 3 genes-14-01305-t003:** The QTL information of the Ear Diameter (ED) trait.

QTL	Chromosome	Position(cM)	LOD	Mapping Interval	Additive_Effect	R^2^
*qED1-1*	1	135.31	2.80	133–136.5	−0.189	0.071
*qED9-1*	9	91.31	3.56	85.8–94.7	0.127	0.102
*qED9-2*	9	99.71	3.16	94.7–103.7	0.122	0.095

**Table 4 genes-14-01305-t004:** The ED, GY, and GY_MPH of eight parents and their F_1_ hybrids and three commercial hybrids.

Materials	Heterotic Group	ED_line	ED_F1	ED_MPH	GY_line	GY_F1	GY_MPH
CML312	nonReid	4.5	5.8	0.47	4.1	12.0	2.20
TRL02	nonReid	4.2	5.5	0.45	4.3	12.7	2.43
CML395	nonReid	4.4	5.7	0.46	4.2	11.1	1.92
CML373	nonReid	4.3	5.4	0.40	4.9	8.8	1.12
Y32	Suwan1	4.6	6.2	0.55	3.7	12.3	2.46
D39	Suwan1	3.8	5.4	0.50	3.9	12.5	2.42
Q11	Reid	4.2	5.5	0.45	3.6	9.7	1.77
Ye107	Reid	3.4			3.4		
Yunrui88 (TRL02×Ye107)	nonReid×Reid		5.5	0.45		12.7	2.43
Dedan5 (D39×Ye107)	Suwan1×Reid		5.4	0.50		12.5	2.42
Yunrui62 (TRL02×D39)	Suwan1×nonReid		5.6	0.40		14.2	2.38

## Data Availability

The GBS sequencing data were deposited in the Genome Sequence Archive (https://bigd.big.ac.cn/gsa) under the accession code PRJCA009949 (https://ngdc.cncb.ac.cn/bioproject/browse/PRJCA009949).

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
