# Peer review of "Identification of Candidate QTLs and Genes for Ear Diameter by Multi-Parent Population in Maize"

_genes, 2023, doi:10.3390/genes14061305_

Round 1
Reviewer 1 Report
In this study, Jiang et al. used a multi-parent population to identify the maize ear diameter (ED) related QTLs, SNP, and associated genes. The authors identified a candidate callose synthase gene associated with ED trait. This is an excellent study showing how quantitative genetic approaches can aid in yield traits studies. I have a few comments listed below to improve the manuscript's quality.
Major:
1. It’s better to check the gene expression of Zm00001d030559 in immature ear tissue of both haplotypes to see if there is any difference correlated with the difference in ED.
2. There is no comparison of current study results with previous studies, especially the QTLs. As the author pointed out in lines 415 to 417, ED has a positive correlation with KRN. Have the authors compared ED results with published KRN results?
3. Figure 2B is confusing. Which environment was the data collected? Why there is no error bar? Any statistical analysis was done to determine if they indeed have significant differences as claimed in lines 244-246?
4. Line 286, why choosing Ye107/Y32 subpopulation needs to be clarified.
Minor:
The authors need to improve their writing quality. There are numerous grammar, and formatting issues. For example,
line 92 the QTL detected by the only using segregating populations used.
line 170 duplicated statements as line 167-168.
line 253 p value is confusing.
Formatting issue and missing punctuation mark:
line 106, 109,
line 110, RIL abbreviation shall be specified earlier in line 80.
line 114, C/N abbreviation
line 115, EU-NAM.
line 169, extra :
line 339, Tother
Reference:
line 124, is #30 reference correct for your statement?
The authors need to improve their writing quality. There are numerous grammar, and formatting issues. For example,
line 92 the QTL detected by the only using segregating populations used.
line 170 duplicated statements as line 167-168.
line 253 p value is confusing.
Formatting issue and missing punctuation mark:
line 106, 109,
line 110, RIL abbreviation shall be specified earlier in line 80.
line 114, C/N abbreviation
line 115, EU-NAM.
line 169, extra :
line 339, Tother
Author Response
Dear reviewer,
I have responded to your comments about our manuscript point by point, Please see the attachment. Hope to get your positive recognition.
Best regards,
Fuyan Jiang

Reviewer 2 Report
Major comments:
1. English is often so poor as to prevent understanding.
2. Models are specified incorrectly.
a. Line 176-180: Missing indices and/or indication of vectors. Which terms are random vs fixed? “Rep” is present in the model but is not described in the methods.
b. Line 209-217: The model described is not MLM (see Yu 2006). In addition, this model is also incorrectly specified, again missing indices and/or indication of vectors.
3. GWAS thresholds (Lines 218-219) are inappropriate and extremely permissive, rendering the majority of this paper’s results and discussion unsupported.
a. The threshold was calculated incorrectly given the stated method: -log10(1/SNP numbers) = -log10(1/264694) = 5.4, not 4.5 as stated in line 218.
b. The method of calculating the significance threshold is not justified and is very permissive. A typical multiple-correction threshold method (Bonferroni) would indicate a threshold of -log10(0.05/# markers) = 6.7, which is much more stringent than the authors’ chosen threshold. While there are other methods that may be used that are somewhat less stringent than Bonferroni (e.g., FDR), this shows that the authors’ threshold is inappropriately lax.
c. To improve: correctly calculate an accepted multiple test correction threshold and re-analyze all results. When this is fixed, I expect that the authors will have no significant results remaining and therefore no paper.
Other comments:
1. Citation issues in the introduction.
a. Line 73-74 cite reference [1], which does not contain the claim made in these lines.
b. Line 124 only cites one reference, but the text states multiple studies were published to date.
2. Repeated issues with describing the F7RILs – e.g., line 131, there are 7 families, not 1,215 families.
3. Methods, results, and discussion repeatedly reference a “three heterotic groups” proposal, but only two of those three (Suwan1xReid and Reid x nonReid) are actually present in the population under study.
4. How was 50kb established as the LD area? Line 220-222
5. Why was only one RIL family in one location examined with QTL mapping (Line 225-234 and 286-314)?
6. Line 228-232: Clarify how these two different LOD thresholds were calculated.
7. Line 240-241: These lines of code are not useful without the input files. Please provide the files.
8. Line 248: CV analysis was not described in methods.
9. Fig 2A: What do the three ears shown for each RIL family represent?
10. Fig 2B-D: Illegible. Also, should be consistent – show all bar plots or all box and whisker plots.
11. Line 266-267: Explain – what significant differences (e.g, mean, SD,…)? Not explicitly supported in results, figures, or tables.
12. Table 2:
a. The columns start, end, and SNP are identical and therefore redundant.
b. What is the “peak_val” column? Appears to be -log(P) but this is not stated.
13. Fig 3: Again, illegible. Also, GWAS on BLUPs is not shown, though is referenced later.
14. Line 295: Explain this linkage disequilibrium block analysis in methods.
15. Line 307: In a biparental mapping population, as used here, the possible average haplotype frequencies are 0%, 50%, and 100% (with of course some variation due to Mendelian sampling) due to the nature of the population, unless there is selection. How are the ratios here (1.75% to 43.6%) occurring?
16. Line 310-311: Here, the authors state that “all families demonstrated similar relationships between the haplotype and ED,” but this is not the case; as can be seen in Fig 4G, some populations show higher ED with AA and others with GG.
17. Fig 4: Again, illegible.
18. Line 352-356: Multiple correlations were examined here without a predetermined hypothesis; I expect that there may be a problem with multiple testing and therefore require a corrected threshold.
19. Table 4: Are these BLUPs? Please clarify.
20. Line 404-419: The body of the paragraph is logically disconnected from its conclusion.
See above.
Author Response
Dear reviewer,
I have responded to your comments about our manuscript point by point, please see the attachment. Hope to get your positive recognition.
Best regards,
Fuyan Jiang

Reviewer 3 Report
In this study, the authors performed GWAS and Linkage Mapping for maize ear diameter (ED) using a large multi-parent population. They identified significant SNPs and QTLs associated with ED, and proposed candidate genes through haplotype analysis. Their results provided important information for maize breeding. Below are some comments and suggestions that the authors need to consider in the revised manuscript.
Figures are blurry.
Line70: replace “grin” with grain
Line 109: Missing punctuation marks
Line 169: Remove colon mark after ED
Line 180: how many replications do you have in each location?
Line 218-219: In line 202, the authors detected 264,694 high-quality SNPs. If so, the threshold should be -5.42 if using the formula - log10(1/SNP numbers). How did you get 4.5?
Line 227, 238: Please provide references for JoinMap 4 and plink
Line 243: I didn’t see any words talk about heterosis.
Line 248-251: why don’t you use BLUP value to evaluate the phenotypes?
How about the heritability of ED?
Line 253: Is it correct for P=0?
Line262-263: Please provide unit in table 1.
Line 265: I suggest including the GWAS results using BLUP value
Line 266-267: Please provide the results from a statistical test.
Line 286-287: why don’t you do the QTL analysis using the data from Baoshan and BLUP value? Why just use data from one RIL population (Ye107/Y32) ?
Line 298-301: The closest gene to SNP_143985532 is Zm00001d030558. How did you exclude this gene? Did you check its expression level? And the expression level of Zm00001d030559 is pretty low. Can you check the expression level in your founder lines in ear primordium?
Line 301: Provide the reference for the expression data source.
Line 303-304: Did you perform the haplotype analysis for other genes in Figure 4C? I wondering these genes are in LD.
Line 313: Typo error, remove one “a”
Line 346: Provide the method for MPH calculation in the method part.
Overall, it's written well.
Author Response

(The authors gave the same response as above.)

Reviewer 4 Report
The authors of this article investigated the genetic basis of ear diameter (ED) in maize and aimed at identifying the quantitative trait loci (QTLs), single nucleotide polymorphisms (SNPs), and candidate genes associated with ED. The authors used a multi-parent population consisting of 1,215 F7 recombinant inbred lines (F7RILs) generated from crosses between an elite maize inbred line and seven elite inbred lines from different heterotic groups. Genome-wide association study (GWAS) and linkage analysis were performed using high-quality SNPs generated through genotyping-by-sequencing. The study identified 11 SNPs significantly associated with ED and revealed three QTLs for ED. The authors also identified a candidate gene, Zm00001d030559, which encodes a callose synthase and showed a positive correlation with ED. The results provide valuable insights into the genetic mechanism of maize ED and offer genetic resources for marker-assisted breeding to enhance maize yield.
Overall, the study addresses an important research question and provides useful findings related to the genetic basis of maize ear diameter. The use of a multi-parent population and the combination of GWAS and linkage analysis strengthen the study design. The identification of significant SNPs, QTLs, and candidate genes adds to the understanding of the molecular mechanisms underlying maize ED.
However, there are a few points that need clarification and improvement in the article:
Section2.1: Parental line selection and representativeness: The rationale behind the selection of the seven elite inbred lines and their representativeness of the three heterotic groups should be further discussed. Providing additional details about the selection criteria and the genetic variations within the heterotic groups would enhance the credibility of the study. Currently, it is still not clear what are the properties of the parents selected apart from the fact that they belong to separate heterotic groups and the team head proposed to do so, which by the way need not be mentioned in a research article and lack credibility without statistical backing.
Why was BLUP chosen instead of BLUEs for the calculation? GWAS typically relies on linear regression models, where BLUEs can properly estimate the effect sizes of genetic variants on the phenotype of interest. It would be helpful to include a statement explaining the rationale behind selecting BLUPs over BLUEs, along with providing the specific R codes used for calculating the BLUPs.
The rationale behind selecting a 50 Kb flanking region of the peak SNP for the candidate gene analysis should be included in the methods section.
More details are required regarding the genotyping-by-sequencing (GBS) process. It would be beneficial to include information about the initial number of SNPs, as well as an explanation of each filtering criterion and how it contributed to reducing the number of SNPs. Additionally, it is important to describe what happens to the dataset at each stage of the pipeline to provide a comprehensive understanding of the GBS workflow.
The choice to focus on subpopulation Y32 × Ye107 and its 162 F7RILs families for QTL mapping raises the question of why all seven populations were not included. It would greatly enhance the value of the article to have an analysis encompassing all populations, especially considering the availability of marker information from them, as utilized in the GWAS. Including this analysis would provide a more comprehensive understanding of the overall findings.
It would be beneficial to include the number of RILs for each population in Figure 2 or consider including a separate table. This addition would provide readers with an exact count of the RILs evaluated from each population, enhancing their understanding of the data presented in the study.
There appears to be a discrepancy between the materials and methods section, where the authors demonstrate BLUP calculations combining environments, and the results section, which only presents GWAS results for each environment independently. To ensure consistency, it is recommended to include the individual environment as well as combined environments statistical analysis in the materials and methods as well as results section, along with the GWAS results obtained through the combined analysis along with individual environments. This addition will provide a comprehensive overview of the findings and align the information presented in both sections.
Functional validation: The study identifies a candidate gene, Zm00001d030559, but no experimental validation or functional analysis of this gene's role in ear diameter is provided. It would be valuable to discuss potential strategies for validating the functional significance of this gene, such as gene expression analysis, knockout experiments, or transgenic approaches.
Comparing the results of the current study with a recent article (https://doi.org/10.1093/gigascience/giac080) with the association studies on numerous maize traits would be beneficial. Specifically, it is worth investigating if any of the genomic regions associated with different phenotypes in their study overlap with the regions identified in the current study of ED. This comparison would provide valuable insights into potential shared genetic factors and the consistency of your findings. Or might give some new insights which will aid in discussing the results of the current study.
Create and cite a GitHub repository to house all the codes used in the study, including R codes for calculating BLUPs, generating figures, and analyzing GBS data. This will allow for easy access and reproducibility of the study's findings.
The article contains several grammatical errors and ill-framed sentences. The authors should carefully proofread the article and improve the overall language and writing style to enhance clarity and readability.
In conclusion, the study presents significant findings regarding the genetic basis of maize ear diameter. However, it requires several improvements as suggested above to enhance the overall quality and impact of the study.
The article contains several grammatical errors and ill-framed sentences. The authors should carefully proofread the article and improve the overall language and writing style to enhance clarity and readability.
Author Response

(The authors gave the same response as above.)

Reviewer 5 Report
The study is conduct by the books to find the genes underlying the phenotypic variation of a trait.
My concern is about the weakness of making it only on one trait. Pretending it is the only/main one to be correlated with grain yield.
There are many other traits involved to establish maize grain yield, considering only the maize ear, you should have widened your study to ear length , number of kernels per ear and thousand kernels weight. And the interaction between these traits is the main purpose of maize breeders. I am sure the phenotypic evaluation you managed on this amazing set of NAM populations was able to collect this data. So the process you showed to identify the genes.
Author Response

(The authors gave the same response as above.)

Round 2
Reviewer 3 Report
I appreciate the authors addressed my questions. I still have some minor suggestions, see below
In the clean version,
Line 226: please including the method for calculating the independent marker.
Line 243: p should be -log10(p). Spaces are required between words and brackets, please check the whole manuscript.
Line 277: The statistical method should be added for the p-value.
For the writing of the value p-value, please be consistent in the manuscript.
The pictures in the revised manuscript is still not clear.
looks good
Author Response
Dear reviewer,
We have responded to your comments about our manuscript point-by-point, Please see the attachment. Hope to get your positive recognition.
Best regards,
Fuyan Jiang

Reviewer 4 Report
The authors have addressed some of the comments, while for most of them, they have explained that those will be addressed in the future or have been addressed elsewhere. All of those points or explanations need to be included in the article. The majority of the suggestions made were straightforward and easily implementable in this article. For example:
Regarding point 3: It is recommended to plot and include an LD plot using the GBS data obtained in this research to support the methods and provide relevant descriptions.
The authors have responded to the reviewer's additional points about the GBS workflow; however, those points need to be incorporated into the manuscript.
The response for point 5 must be included in the manuscript to clarify to the readers that only one population is discussed, so that readers do not waste time searching for results associated with other crosses.
For point 7: The model used for BLUP calculation across all environments has been provided, but the model used to analyze individual environment datasets needs to be included as well in the manuscript.
Point 8: The authors should discuss or include a couple of points about the future prospects of the research in the discussion or conclusion section.
Point 9: This can be easily accomplished and must be included in the manuscript to provide an overall idea if any other traits are associated in an antagonistic manner with the ED. The readers need to have a comprehensive understanding of the work conducted and how to approach a given phenotype and associated research.
Point 10: If not already done, cite the GitHub repository or create one. It is important to make the specific analysis and plotting codes used in this manuscript publicly available for better reproducibility and to ensure the analysis was performed correctly.
There still are some grammatical errors
Author Response

(The authors gave the same response as above.)

Reviewer 5 Report
I would like to express my appreciation for your response to my comment. It reinforces the notion that you have performed an exceptional analysis of the various aspects contributing to the green yield of maize, encompassing six traits, if my assessment is accurate.
However, I hold the view that the division of each trait's analysis into separate papers is not advisable. This approach overlooks the crucial interactions and balance between these traits, which are of paramount importance to corn breeders.
If your objective is to concentrate on gene function, I suggest that your paper should delve deeper into the proof of concept. This could be achieved through the evaluation of Crispr/cas9 mutants or NILs sister lines within your heterotic pattern, further substantiating the significance of your candidate genes.
Author Response

(The authors gave the same response as above.)

Round 3
Reviewer 4 Report
The authors have addressed most of my points
There are a few minor grammatical errors.